# Application of C5.0 Algorithm for the Assessment of Perceived Stress in Healthcare Professionals Attending COVID-19

**DOI:** 10.3390/brainsci13030513

**Published:** 2023-03-20

**Authors:** Juan Luis Delgado-Gallegos, Gener Avilés-Rodriguez, Gerardo R. Padilla-Rivas, María De los Ángeles Cosío-León, Héctor Franco-Villareal, Juan Iván Nieto-Hipólito, Juan de Dios Sánchez López, Erika Zuñiga-Violante, Jose Francisco Islas, Gerardo Salvador Romo-Cardenas

**Affiliations:** 1Departamento de Bioquímica y Medicina Molecular, Facultad de Medicina, Universidad Autónoma de Nuevo León, Monterrey 64260, Mexico; 2Escuela de Ciencias de la Salud, Universidad Autónoma de Baja California, Ensenada 22890, Mexico; 3Universidad Politécnica de Pachuca, Carretera, Carretera Ciudad Sahagún-Pachuca Km. 20, Ex-Hacienda de Santa Bárbara, Zempoala 43830, Mexico; 4Althian Clinical Research, Calle Capitán Aguilar Sur 669, Col. Obispado, Monterrey 64060, Mexico; 5Facultad de Ingeniería, Arquitectura y Diseño, Universidad Autónoma de Baja California, Carr. Transpeninsular 391, Ensenada 22860, Mexico

**Keywords:** decision tree, COVID-19 stress, healthcare professionals in Mexico, explainable artificial intelligence for healthcare

## Abstract

Coronavirus disease (COVID-19) represents one of the greatest challenges to public health in modern history. As the disease continues to spread globally, medical and allied healthcare professionals have become one of the most affected sectors. Stress and anxiety are indirect effects of the COVID-19 pandemic. Therefore, it is paramount to understand and categorize their perceived levels of stress, as it can be a detonating factor leading to mental illness. Here, we propose a computer-based method to better understand stress in healthcare workers facing COVID-19 at the beginning of the pandemic. We based our study on a representative sample of healthcare professionals attending to COVID-19 patients in the northeast region of Mexico, at the beginning of the pandemic. We used a machine learning classification algorithm to obtain a visualization model to analyze perceived stress. The C5.0 decision tree algorithm was used to study datasets. We carried out an initial preprocessing statistical analysis for a group of 101 participants. We performed chi-square tests for all questions, individually, in order to validate stress level calculation (*p* < 0.05) and a calculated Cronbach’s alpha of 0.94 and McDonald’s omega of 0.95, demonstrating good internal consistency in the dataset. The obtained model failed to classify only 6 out of the 101, missing two cases for mild, three for moderate and one for severe (accuracy of 94.1%). We performed statistical correlation analysis to ensure integrity of the method. In addition, based on the decision tree model, we concluded that severe stress cases can be related mostly to high levels of xenophobia and compulsive stress. Thus, showing that applied machine learning algorithms represent valuable tools in the assessment of perceived stress, which can potentially be adapted to other areas of the medical field.

## 1. Introduction

With the global spread of the COVID-19, both medical and allied healthcare professionals have become the most highly affected sectors [1,2,3]. In developing democracies, the public health system became engulfed by the overwhelming levels of stress [4,5]. In addition, the situation becomes even more taxing for attending personnel as they not only deal with the burdened system [6] but also with the enemy (COVID-19) upfront. It is here, where they can also become prey to the disease [7]. Recently in Mexico, reports for the period of late February to 23 August showed that over 97,600 healthcare professionals had become infected with COVID-19 [8]. Hence, Mexico showed atop of all Latin America countries in infection-to-death rate (>10%) [9]. The number of “total confirmed”, possible, active cases, and mortality of COVID-19 amongst physicians, almost doubled during the period of 16 August up to 3 November, potentially generating high levels of stress on them. This is of particular interest when we consider stress as a potential trigger to lose focus during procedures or while attending to patients; therefore, enabling conditions for COVID-19 infection, or making costly mistakes [10]. 

According to the Pan American Health Organization (PAHO), Mexico has the highest number of healthcare workers infected with COVID-19 in Latin America [11]. In 28 December 2020, the number of health care professionals affected by COVID-19, as reported by the National health ministry, was just over 182,200 [12]. Reports show that both physicians and nurses have similar levels of burnout and emotional fatigue [3,13,14,15]. Physicians typically work in a more independent manner. This, along with their long shift hours, high-sense of duty, work ethics, and the fact that they partake in multiple jobs normally of low wages, becomes a source of additional stress [8]. With the data being generated while facing the disease, it is important the apply rapid methods that allow study of this scenario and allow development of policies or strategies. For this purpose, machine learning algorithms have proved efficient in the analysis of stress in working employees [16,17,18]. Still, for medical applications, it is important for the algorithm to provide explainability for computer-aided diagnosis [19]. Therefore, in this study, we propose the use of the C5.0 algorithm to assess perceived stress in healthcare workers exposed to COVID-19, generating an explainable classification diagram that contributes to the understanding of mental health in pandemic scenarios. 

Recent developments in computational modeling have led to the ever-evolving field of artificial intelligence, which, when combined with neuro- and behavioral science, has created the new field of computational psychiatry [20,21]. Computational psychiatry helps to model and understand underlying mental illness, allowing the prediction of potential behavioral patterns, improving classification, and assisting the physician to provide a faster and personalized medical attention [22]. Nowadays, machine learning algorithms are promising technologies used by various healthcare providers, as they result in better scale-up, speed-up, processing power, and reliability, which translates into a more efficient performance of the clinical team [23,24,25]. Therefore, a trend is to use these techniques to better understand, and fight against the current pandemic and other chronic diseases, especially when the resulting model could have a graphical-based explanation [26]. Using well-known machine learning algorithms, such as decision trees, for establishing classification systems are but one of the many features of their application [27]. Typically, it is possible to classify a population into branch-like segments that generate an inverted tree [27,28]. These algorithms can efficiently deal with large, complicated datasets without imposing a complicated parametric structure [28]. Researchers have reported the use of these types of algorithms for applications in the study of behavioral and mental health [29] and on the use of computational based methods to classify stress from data generated by sensor devices [30]. Thus, it is possible to use these tools to better understand disease and propose different clinical paths, and to classify subgroups of patients for different diagnostic tests, treatment strategies, and assessment of mental health-related conditions [31,32].

Several approaches on machine learning-based stress assessments have been reported. A common method considers the use of bio-marker data to stratify stress on several levels [33,34], since these algorithms are able to asses not only stress, but depression and anxiety as well [18,35]. For COVID-19-related stress evaluation, the use of these type of algorithms has been previously explored for general population studies, based on distributed questionnaires data [36,37], which allows for exploration of data acquired from these kinds of questionnaires for clinical applications [38]. 

Specifically, decision tree algorithms have been able to obtain 92% accuracy, providing not only a reliable stress categorization [39], but they also generate a visual model that allows to analysis of the actual scenario of the problem, which is not common with machine learning algorithms. 

In machine learning, a common strategy used for data analytics is the cross-industry standard process for data mining method (CRISP-DM). This method defines six steps for data-based knowledge projects. This strategy begins with defining problems and objectives (business understanding), followed by data insights (data understanding). Next, defining a dataset and its analysis (data preparation), and results from this analysis generates a model (modeling). Once generated, it is evaluated (evaluation), and if the goal is achieved, it can be implemented [40].

Given that it is possible to use decision tree algorithms to identify prominent features that influence stress [16], it is feasible to apply this type of algorithms to obtain an explicative model of the studied scenario. Additionally, the proven efficiency of the C5.0 algorithm as a biomedical decision support tool for assisted diagnosis makes it a likely tool for the case [41,42,43]. In the current work, we studied the application of a C5.0 decision tree algorithm, as proposed in the literature, to locate the combination of factors needed to classify, correctly, healthcare professionals attending to COVID-19 patients, by the category of perceived stress. This provides a graphical tool that allows a better understanding of the mental health of healthcare professionals at the beginning of the COVID-19 pandemic in northeast Mexico.

## 2. Materials and Methods

Some other work on stress perception during the COVID-19 pandemic has been reported regarding healthcare workers [44,45]. Our work is based on previously reported adapted COVID-19 stress scale (ACSS) data [1], at the beginning of the pandemic, in healthcare workers in northeast Mexico. The dataset was previously classified into different categories of perceived stress for healthcare professionals attending to COVID-19 patients: five variables were defined (danger and contamination, xenophobia, traumatic stress, compulsive checking, and social economical) and four different results were defined, with scores per area: 0–6 absent, 7–23 mild, 13–18, moderate, 19–24 severe. A total tallied score of all the areas was obtained, and further analyzed and correlated in accordance to job-specific characteristics [1]. We adapted the analysis method using the CRISP-DM model, commonly used in data analytics [40], having the same number of stages and sequences, as shown in Figure 1. Initially, we performed a data structure study from the data analytics scope to consider the type of variables from the ACSS. This was to establish the type of variables and how they contributed to the context of the ACSS, along with the four categories of stress defined from the scores as outcomes: absent, mild, moderate and severe. 

Next, we performed a data validation analysis considering statistical tests to confirm relations between variables from the scales and classification outcomes from the raw data, and to confirm internal consistency [46]. This was completed by obtaining both Cronbach’s alpha and McDonald’s omega from the raw ACSS responses and a Pearson chi-square statistic applied to the ACSS and the resulting stress scale. We followed the validation process with a data distribution analysis to study stress components for model selection and interpretation. This measured the central tendency of the professional profile, which included the profession and work area from the healthcare workers who participated in the study, as well as for the ACSS and the resulting stress class.

Given that the approach of this work is to provide an AI-based method that could become a tool for clinical decision making, we selected a decision tree (DT) model to study the relations and classification routes for stress level according to data from its respective scales. 

We carried out an accuracy analysis based on the results from algorithm training, as well as a sensitivity and specificity analysis by splitting the categories defined for stress into different subgroups for healthy and disease states.

### 2.1. Descriptive Statistical Analysis

We performed both the statistical and algorithm performance analysis in R language to obtain behavior patterns and understanding of data distribution. For data preparation and preprocessing, we also carried out a descriptive statistical study to understand data structure and distribution. To obtain valuable information for model interpretation, measures of central tendency were obtained from the professional profile data of the healthcare workers who participated in the study, as well as from the ACSS. 

For the instrument validation purposes, we estimated the value of Cronbach’s alpha considering the numerical values from all participant responses [47]. Finally, we applied Pearson chi-square statistics using SPSS (ver. 21) to the ACSS areas to show results robustness [48].

### 2.2. Application of C5.0 Algorithm 

Following the statistical analysis on the instrument results, we developed a DT to behave as a computational supportive scaffold for the study of mental illness. We opted to use a C5.0 algorithm to analyze and classify the stress level from the dataset and for construction of a classification tree, as used in previous health-related scenarios [49]. This algorithm uses information gain as its splitting criteria and the binomial confidence limit method for the pruning technique, improving the feature selection and reducing error pruning. These methods have been reported useful to build efficient classifying models having small datasets, given the mathematical background of the model [50]. Additonally, DT outperforms other algorithms with smaller datasets, as in this case.

Following both the statistical and computational analysis of the instrument and dataset, we analyzed the performance based on sensitivity and accuracy on the generated model [51]. Given the size of the dataset, the confusion matrix obtained from the algorithm training was used to define the accuracy of the obtained model. Then, sensitivity and specificity calculations were completed using the results of the confusion matrix. Given that there are four different levels of stress defined as outcome, three different combination subgroups were used to define healthy and disease states. Conclusions were drawn from the results of the analysis, as well as routes defined by the tree model branches considering initial statistical analysis. The application of this algorithm is not intended as a classification tool but as a computer-aided tool that provides a wider scope of stress in healthcare workers. For this, the whole dataset was used to train the algorithm and to obtain the DT with the use of R and RStudio.

### 2.3. Dataset

As mentioned before, the study considers a dataset obtained from 106 participants from which information related to medical or healthcare education, work field and experience. Then, the data is built into a stress concept conformed by five components, which are: danger + fear of contamination, socioeconomical, xenophobia, traumatic stress and compulsive checking. Danger + fear of contamination refers to perceived stress related to the probability of being exposed and contracting the disease. The socioeconomical factor refers to financial-related stress that is associated with the chance of losing their job and the financial burden of becoming unemployed. Xenophobia is a scale that refers to the fact that the disease comes from abroad and it might not be possible to stop it. Traumatic stress refers to the emotional burden related to work with COVID-19 patients, and compulsive checking it related to compulsive behavior around the need to look for information about the disease.

## 3. Results

We applied an initial preprocessing statistical analysis to the 106 entries dataset. After eliminating missing data entries for statistical and algorithm-based analysis preparation, we used a group of 101 entries for the study. Besides explainability from the graphical output, decision trees have proved useful for small datasets [52]. Still, the dataset is greater than the minimal size of 62 required for decision tree models [50]. 

From the total entries, we counted the frequency of the profession and work area variables, as shown in Table 1.

We then built upon the five areas of the ACSS, calculating the central tendency metrics for each of these components based on the cumulative result of each participant, as shown in Table 2. 

Given that we based each feature on the addition of the responses from the survey, we considered all the values from each question and participant for the calculation of Cronbach’s alpha, which shows a good internal consistency (0.94) for the whole survey instrument and data, and a similar result for McDonald’s omega (0.95). In addition, Appendix A shows chi-square tests to each question in order to define significance in the relationship of the variables. Table 3 shows the result of the test for each scale area and each question, and for the cumulative ACSS.

Both results from Cronbach’s alpha and the chi-square test show internal consistency of the data and validate the dependence for stress level calculation, ensuring the dataset quality for algorithm-based analysis. Distribution for the stress level classification in healthcare personnel calculated from the ACSS is shown in Figure 2.

Figure 3 shows a scatter plot from the intersection from the xenophobia and danger + fear of contamination scales from the ACSS, allowing to observe the distribution of the stress levels based on these two variables in some areas of the graph.

Stress scale distribution showed in Figure 2 shows the general incidence of the stress level in healthcare professionals at the beginning of the pandemic. Although imbalanced, commonly in medical data, correlation distribution showed in Figure 3 confirms the feasibility to use the dataset, despite the size and imbalance, for the purpose to decipher medical context [53].

Following the descriptive statistical analysis, we trained a decision tree model with the preprocessed dataset (*n* = 101) using the C5.0 algorithm [28,49], considering the stress level to be the target variable. We used all areas of the ACSS including participant profession and work area as the predictive variables to find any relationship between them to predict stress level. Figure 4 shows the decision tree obtained from the dataset. 

At the class level, a set of boxes with all four levels of stress are observed. In each box, the extreme right bar corresponds to the severe level indicator, followed, to the left, by moderate, mild and absent levels, respectively. Despite declaring the features related to the participant profession and work area, these variables did not provide valuable information gain to be considered in the model. Table 4 shows the confusion matrix from the obtained decision tree model, where only 6 out of 101 entries were incorrectly classified, missing two cases for mild level, three for moderate and one for severe. All these errors were classified only in neighboring levels, giving the model an accuracy of 94.1%.

To analyze model performance, a sensitivity and specificity calculation were carried out. For these, three different scenarios were considered based on the stress classification outcome from the dataset, dividing entries into healthy and disease groups. Calculation was completed with the figures from the confusion matrix. Results are shown in Table 5.

## 4. Discussion

Our purpose was to define a statistical and computational framework algorithm to analyze and understand stress levels in healthcare professionals for the impact of the COVID-19 pandemic and to potentially define a graphical self-explainable clinical tool, which can be further used as a severity predictor of stress.

A dataset related to the ACSS, as defined by Delgado-Gallegos et al., was studied with a calculated Cronbach’s alpha of 0.94, which shows a good internal consistency; stress levels were calculated as a geometrical result from the addition of five scales from the survey defined as danger + fear of contamination, socioeconomic stress, xenophobia, traumatic stress and compulsive checking. Chi-square tests were carried out for all questions individually, looking to validate stress level calculation. Statistical significance (*p* < 0.05) was found in most of the questions, considering the answers of all participants, except one question for the traumatic scale, and four for the compulsive checking scale (all shown in Appendix A). However, all scales showed statistical significance when the test was applied to the accumulated value for each of these scales, as seen in Table 3; thus, validating, the use of the ACSS in a population [1,54]. Therefore, the use of this model can be re-adapted to help in correctly assessing and providing a faster diagnosis and opportune treatment.

From the central tendency metrics statistical analysis, no relation was observed between participant profession and work area, similar analysis was done for the stress scales which showed an exception for danger + fear of contamination joint scale, all other areas had a similar maximum value but with different means. Therefore, considering the results from the preprocessing stage, the dataset shows good quality, independence, and internal consistency for algorithm analysis. All 101 entries from the dataset were used to train a decision tree model by the C5.0 algorithm, where stress level was defined as the target variable, with participant profession, work area, and cumulative stress scales as predictors. The resulting model showed an accuracy of 94%, adding a more precise assessment to the initial stress classification. Nonetheless, the algorithm did not find enough information gain from the participant profession, work area, and the socioeconomic scale. Neglecting these variables from the resulting model allows to understand that experience and day-to-day work routine are not a factor on how healthcare professionals perceive stress. Resilience could help explain this pattern, as it is an adaptation mechanism in which a person, overtime, can handle stress in overwhelming situations [15,55].

Computational psychiatry states the similarity between the brain and a computer and proposes the use of computational terminology for the study of mental illness [56]. Our results show interesting data denoting hypothetical tendencies based on the purity of the resulting branches of the decision tree, where severe stress cases can be related mostly to high levels of xenophobia and compulsive stress, as shown by the relation of the threshold values from the extreme right route of the decision tree, which are above the 3rd quartile for xenophobia and compulsive stress scales, and from the measures of central tendency shown in Table 2. In a similar manner, absent stress level comes from the scenario of combined thresholds below the 1st quartile from xenophobic, compulsive and traumatic stress scales. It is interesting to note that the danger + fear of contamination scale can be used to find both mild and moderate cases, despite being a larger joint scale.

Even though there are various classification algorithms, such as K-Nearest Neighbors, Support Vector Machines, Naive Bayes, Random Forest, Radial Basis Function or Adaptive Boosting (AdaBoost) that are used for classification process with prominent accuracy and performance, it has been previously reported that with the use of decision tree algorithms, it is possible to rely on a few variables from a health-related problem to stratify patients with a visual tool that empowers clinical decision [41,42,43]. Given the size of our dataset (*n* = 101), this constitutes an efficient input for the C5.0 algorithm, which was further confirmed with the sensitivity and specificity analysis. In addition, the sensitivity and specificity analysis showed acceptable results despite the few severe stress cases. Appendix A shows the analysis of the studied dataset with the algorithms mentioned above.

Currently, machine learning and decision tree algorithms are still in their initial stages of application in the medical field. Recently, Yu et al. published a retrospective study on the conditions to predict metabolic syndrome [57] and Peng et al. published a recent study on the prediction of exacerbation of chronic obstructive pulmonary disease using key indicators, the result had an overall accuracy of 80.3% with a confidence level of 95% [58]. Machine learning has also been used to identify complex patterns in emergency hospital services, which implies intelligence data-driven decisions even under overwhelming circumstances [59].

## 5. Conclusions

This is only a fist approximation based on recent data from healthcare professionals in the northeast part of Mexico [1] and the first study of its kind using the C5.0 decision tree algorithm model on the assessment of stress on self-explainable model basis. Because of the mathematical foundation of these algorithms, it allows not only to obtain a better understanding of a problem, but also to generate accurate predictions. The need of larger datasets and machine learning methodological approaches is well established. Therefore, the impact of applying machine learning algorithms represents a window of opportunity in actual global health and in the decision-making process of developing health policies, based on large-scale studies. For clinical decision-making scenarios, decision trees are specifically useful to simplify assisted diagnosis given the ease of understanding, expanding the scope of computer assisted diagnosis.

This work contributes to mathematical-informed understanding of mental illness and computational psychiatry; thus, forming a diagnostic tool to help in the assessment of patients. In this study, we analyzed healthcare professionals’ answers, as they are one of the most affected sectors in the pandemic [60]. In addition, an expansion of this method with the use of algorithm combinations could provide efficient clinical-assisted tools that could apply to scenarios of the internet of medical things; real-time measurement of compounds or metabolites could be analyzed to decipher medical context, as in this work, or even to reach customizable medicine. In addition to uses from the COVID-19 pandemic, it can be used to understand different stress factors and how they can interfere with performance and the social dynamics in different populations.

## 6. Limitation

The main goal of this work is to show that the mathematical-/computer-based analysis applied to a very specific population allowed to identify patterns in behavior and mental health, despite the fact that the sample size could not be big enough for a formal data analytics study. Applying synthetic methods to increase sample size or to balance the target variable could affect the actual scenario of the data from the population analyzed during a small and very specific period of time, making the founded patterns meaningless. The use of a decision tree to the diagnosed population during the COVID-19 pandemic contributes to the understanding of mental health and behavioral patterns within an emblematic event in human history.

A formal analytics study was added as Appendix A. The application for computer-aided diagnosis is suggested for future work.

## Figures and Tables

**Figure 1 brainsci-13-00513-f001:**
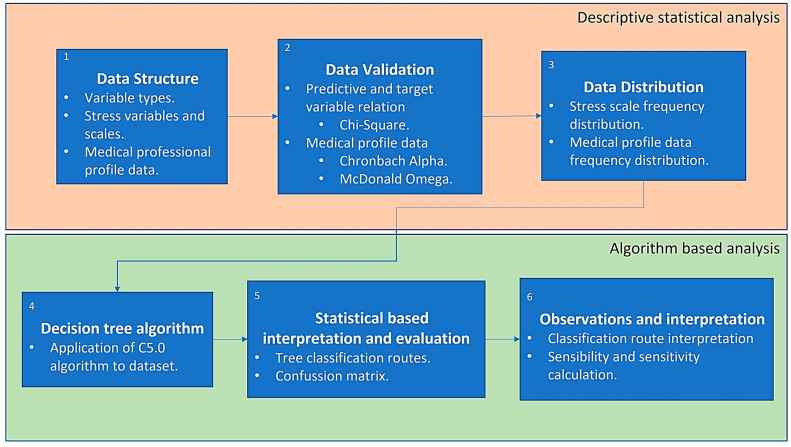
Methods for machine learning-based analysis on stress scales of healthcare workers.

**Figure 2 brainsci-13-00513-f002:**
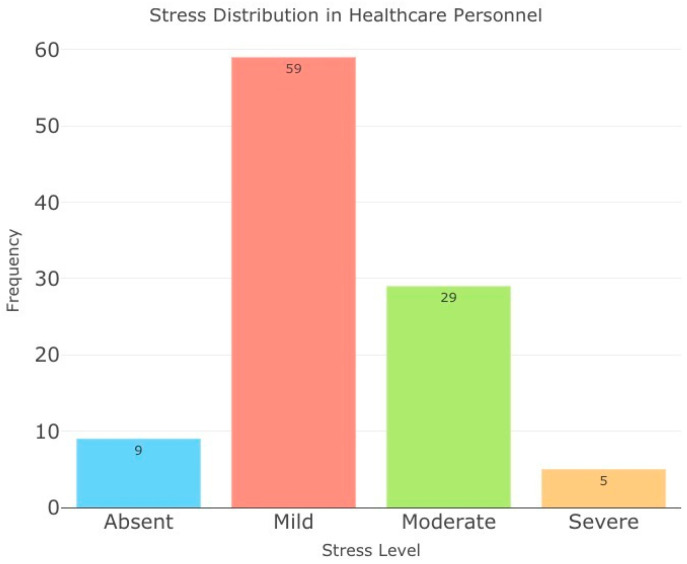
Stress level distribution in healthcare personnel. (Left to right) Absent, Mild, Moderate, Severe.

**Figure 3 brainsci-13-00513-f003:**
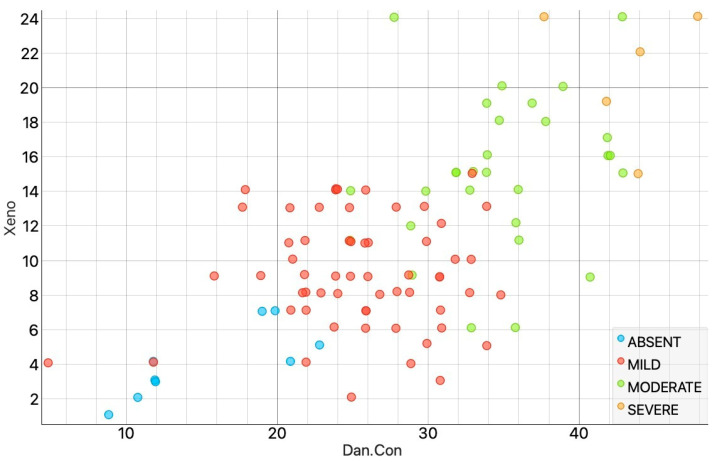
Stress level distribution in healthcare personnel from the intersection of xenophobia and danger + fear of contamination scales from the ACSS.

**Figure 4 brainsci-13-00513-f004:**
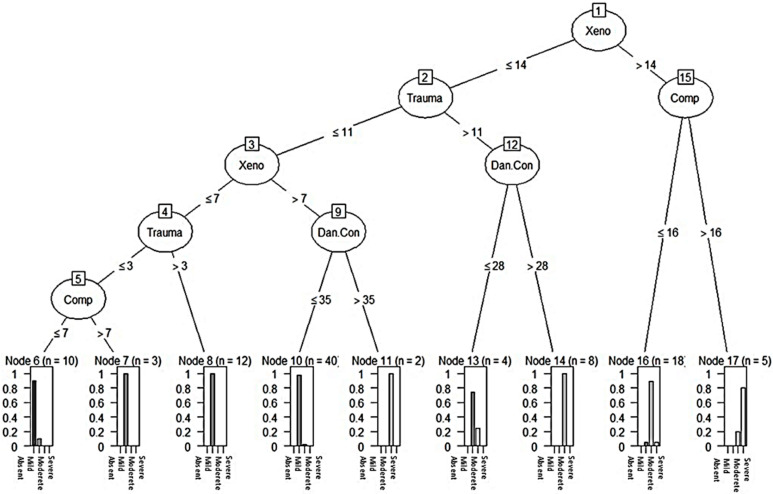
Decision tree applied into healthcare personnel stress scale level dataset. Atop variables influencing stress are xenophobia (Xeno) and compulsive checking (Comp), which leads to severe stress. Traumatic stress (Trauma) and danger + contamination (Dan Con) also influenced the perception of stress. The socioeconomical variable did not influence the outcome of the decision tree.

**Table 1 brainsci-13-00513-t001:** Frequency count on participant profession and work area.

Participant Profession	Counts	Participant Work Area	Counts
Medical Student	2	Front line health professional	29
Nursing Staff	10	Others	34
Physician	69	COVID-19 designated area	11
Physician in community service *	4	Surgical	11
Resident	15	ER	9
Technician	2	Internal medicine	8

* Physician in community service. A medical student who has finished the required medical school training in Mexico and is doing a compulsory one-year internship at a local community hospital or healthcare facility.

**Table 2 brainsci-13-00513-t002:** Central tendency metrics for the adapted COVID-19 stress scale features.

Stress Scale Feature	Min	1st Quartile	Median	Mean	3rd Quartile	Max
Danger + fear of contamination	5	23	25	25.2	33.75	48
Socioeconomical	4	14	17	16.27	19	24
Xenophobia	1	7	10.5	10.9	14	24
Traumatic stress	0	2	6	7.37	12	22
Compulsive checking	0	5	8	9.38	13.75	24

**Table 3 brainsci-13-00513-t003:** Analysis per general area.

COVID Areas	Absent	Mild	Moderate	Severe	Xi2	Sig
Danger + fear of contamination	3	23	58	17	64.98	<0.001
Socioeconomical	30	35	24	12	11.673	<0.009
Xenophobia	15	45	29	12	27.119	<0.001
Traumatic stress	47	25	21	8	31.238	<0.001
Compulsive checking	26	43	22	10	22.129	<0.001
CSS general score	9	59	28	5	72.109	<0.001
Xi2 = chi-square test						

**Table 4 brainsci-13-00513-t004:** Confusion matrix of obtained decision tree model for stress level classification.

Classified as				
(a)	(b)	(c)	(d)	Actual Class
9				(a) Absent
1	57	1		(b) Mild
	2	26	1	(c) Moderate
		1	4	(d) Severe

**Table 5 brainsci-13-00513-t005:** Decision tree sensitivity and specificity calculation from stress scales dataset.

	Healthy: Absent + Mild + Moderate Disease: Severe	Healthy: Absent + Mild Disease: Moderate + Severe	Healthy: Absent Disease: Mild + Moderate + Severe
Sensitivity	0.8	0.91	0.989
Specificity	0.989	0.98	0.9

## Data Availability

Dataset may be downloaded from Kaggle (https://www.kaggle.com/chepox/css-mexico, accessed on 19 January 2021). Code may be accessed from github (https://github.com/Bio-Math/COVID19-Stress-Health-Professionals) (19 January 2021).

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
