# Peer review of "Application of C5.0 Algorithm for the Assessment of Perceived Stress in Healthcare Professionals Attending COVID-19"

_brainsci, 2023, doi:10.3390/brainsci13030513_

Round 1
Reviewer 1 Report
The topic of this paper is interesting and need major corrections which are as follows:
1- The main contribution of this work is not clearly explained in the current version, the authors must explain it in more detail.
2- Subsection "1.1 Background" is only one subsection, that should be involved in the Introduction section without a subsection. Or make it in the new section.
3- Related work is missing.
4- Materials and Methods section, the current version is not clear and makes the readers In the confusion that the authors must add a figure to explain the proposed method in detail to follow their ideas.
Author Response
We appreciate the feedback from the reviewer. We hope that the
1- The main contribution of this work is not clearly explained in the current version, the authors must explain it in more detail.
-We added the following sentence to the abstract:
"Here we propose a computer-based method to better understand stress in healthcare workers facing COVID-19 at the begining of the pandemic"
2- Subsection "1.1 Background" is only one subsection, that should be involved in the Introduction section without a subsection. Or make it in the new section.
- We eliminated the subsection
3- Related work is missing.
- We added the following references:
- M. Cutillo et al., “Machine intelligence in healthcare—perspectives on trustworthiness, explainability, usability, and transparency,” NPJ Digit. Med., vol. 3, no. 1, p. 47, 2020.
[25] K. A. Bhavsar, A. Abugabah, J. Singla, A. A. AlZubi, and A. K. Bashir, “A comprehensive review on medical diagnosis using machine learning,” Comput. Mater. Contin., vol. 67, no. 2, p. 1997, 2021.
[26] A. J. London, “Artificial intelligence and black‐box medical decisions: accuracy versus explainability,” Hastings Cent. Rep., vol. 49, no. 1, pp. 15–21, 2019.
[42] B. Bareeqa et al., “Prevalence of depression, anxiety and stress in china during COVID-19 pandemic: A systematic review with meta-analysis,” Int. J. Psychiatry Med., vol. 56, no. 4, pp. 210–227, 2021.
4- Materials and Methods section, the current version is not clear and makes the readers In the confusion that the authors must add a figure to explain the proposed method in detail to follow their ideas.
- Fig. 1 explains the methodology along with the addition of the graphical abstract.
Reviewer 2 Report
Dear Authors
In relation to your work I have found some points that I would like to comment on:
1.- The penultimate paragraph where you explain the use of ACSS and what the values mean, does not seem to make sense in the introduction and if in the material and methods. This is because, from my point of view, it allows future readers to understand how you have constructed your proposed model.
2.- Could you explain why you have used Crombach's alpha and not McDonald's omega?
McDonald's omega uses the weighted sum of the standardized variables and reflects the true level of reliability. In addition, the alpha coefficient is used to work with continuous quantitative variables, this being, perhaps, a problem when dealing with variables whose categorization are Absent, Mild, Moderate, and Severe.
3.- Regarding the participants, could you please indicate from which country they are from?
A situation that is related to the possible use of an ACSS questionnaire is that you have not indicated if it is validated for the type of population that has participated in your study and it is the one you have used to validate your model.
4.- Please, if you include a specific heading for limitations, you should remove the limitations you have included in the discussion.
Author Response
We appreciate the comments from the reviewer. We hope we were able to address them:
1.- The penultimate paragraph where you explain the use of ACSS and what the values mean, does not seem to make sense in the introduction and if in the material and methods. This is because, from my point of view, it allows future readers to understand how you have constructed your proposed model.
- We modified the introduction and Methods section according the reviewer suggestion
2.- Could you explain why you have used Crombach's alpha and not McDonald's omega?
McDonald's omega uses the weighted sum of the standardized variables and reflects the true level of reliability. In addition, the alpha coefficient is used to work with continuous quantitative variables, this being, perhaps, a problem when dealing with variables whose categorization are Absent, Mild, Moderate, and Severe.
- We added the McDonald omega results (0.95) to both the abstract, methods and results section.
3.- Regarding the participants, could you please indicate from which country they are from?
A situation that is related to the possible use of an ACSS questionnaire is that you have not indicated if it is validated for the type of population that has participated in your study and it is the one you have used to validate your model.
- We added the note that healthcare workers were from Northeast Mexico at the abstract, the methods section and it is also noted in the previous report.
J. L. Delgado-Gallegos, R. de J. Montemayor-Garza, G. R. Padilla-Rivas, H. Franco-Villareal, and J. F. Islas, “Prevalence of stress in healthcare professionals during the covid-19 pandemic in Northeast Mexico: a remote, fast survey evaluation, using an adapted covid-19 stress scales,” Int. J. Environ. Res. Public Health, vol. 17, no. 20, p. 7624, 2020.
4.- Please, if you include a specific heading for limitations, you should remove the limitations you have included in the discussion.
- We modified the discussion and conclussion section according reviewer suggestion.
Reviewer 3 Report
Dear authors.
Thank you for the opportunity to review this paper and congratulations on your work.
The manuscript aims to present the application of an algorithm (C5.0) to create decision trees to assess perceived stress in healthcare professionals involved in the care of COVID-19.
The abstract is of adequate size and accurately represents the manuscript.
The keywords could perhaps include a more specific MeSH descriptor in order to improve the scientific search.
The introduction and the theoretical framework are very well explained, with a large number of references that help to introduce and understand the research topic. The part that explains computational modeling, machine learning, and its application in biomedicine is very complete, so non-expert readers can understand this research and continue reading.
The materials and methods section is very complete, with a good explanation of the steps and processes followed, including those related to the application of the algorithm and decision trees.
The Results section presents the findings, accompanied by descriptive analyses and analyses to demonstrate the internal consistency of the data obtained (Cronbach's alpha) and dependence (Chi-square test), which are indicated. The performance of the model is certified with the respective sensitivity and specificity analyses.
The remaining sections present a very good summary of the findings, highlighting their usefulness and showing the limitations encountered during the research. Perhaps a commentary on further lines of work is missing (on the understanding that this, as the authors indicate, is a first approximation to show that the mathematical/computational analysis applied to a very specific population). It is also accompanied by the complementary material necessary to check the traceability of the research.
Congratulations on your work.
Best regards.
Author Response
We appreciate reviewer feedback.
We added the following line to the conclusion according reviewer suggestion
Because of the mathematical foundation of these algorithms that allows not only to obtain a better understanding of a problem, but also to generate accurate predictions. The need of larger datasets and Machine Learning methodological approaches is well established
Round 2
Reviewer 1 Report
The authors have addressed some of my comments, but still there are some comments that need to address carefully, such as
-Materials and Methods section, the current version is not clear and makes the readers In the confusion that the authors must add a figure to explain the proposed method in detail to follow their ideas.
Author Response
We appreciate reviewer comments.
In order to provide more detail on the research methods, we improveg Figure1 as follows:
Reviewer 2 Report
Dear authors
Thanks for including the modifications that improve the quality of your manuscript.
Author Response
Thank you for your comments
Kind regards